# Learning Controllable Elements Oriented Representations for Reinforcement Learning

## Abstract

Deep Reinforcement Learning (deep RL) has been successfully applied to solve various decision-making problems in recent years. However, the observations in many real-world tasks are often high dimensional and include much task-irrelevant information, limiting the applications of RL algorithms. To tackle this problem, we propose LCER, a representation learning method that aims to provide RL algorithms with compact and sufficient descriptions of the original observations. Specifically, LCER trains representations to retain the *controllable elements* of the environment, which can reflect the *action-related* environment dynamics and thus are likely to be task-relevant. We demonstrate the strength of LCER on the DMControl Suite, proving that it can achieve state-of-the-art performance. To the best of our knowledge, LCER is the first representation learning algorithm that enables the pixel-based SAC to outperform state-based SAC on the DMControl 100K benchmark, showing that the obtained representations can match the oracle descriptions (*i.e.* the physical states) of the environment.

## 1 Introduction

Deep Reinforcement Learning (deep RL) has proven its ability to solve difficult sequential decision-making problems such as Dota2 (Berner et al., 2019) and StarCraft (Vinyals et al., 2019). However, it is still a challenge to apply deep RL to many real-world tasks, because the observations in these tasks are often high dimensional (*e.g.* pixels) and include much task-irrelevant noise. To address these issues, representation learning is introduced in deep RL to provide information-dense descriptions of the original observations, and thus reduce the difficulty of solving RL problems.

Many representations learning algorithms are proposed with the design goal to make the representations be predictive of properties of future states. Such a goal is typically achieved by imposing the representations to reflect the dynamics of the environment (Mazoure et al.; Lee et al., 2020b; Schwarzer et al., 2020; Gelada et al., 2019). However, the dynamics of the environment are not always task-relevant(*e.g.* the noisy and varying background in control tasks). This can be problematic because those task-irrelevant dynamics will entice the representations into encoding task-irrelevant information(*e.g.* the noisy background). Since the capacity of representations is limited, the obtained representations can miss important task-relevant information (Gelada et al., 2019) or encode too much unimportant task-irrelevant information (Zhang et al., 2020). Therefore, it is essential to learn representations with more task-relevant environmental dynamics.

We envision that action-related dynamics are likely to be more task-relevant because actions are the only feedback from the agent to the environment. For example, in control tasks and robotics particularly, the agent can complete its task only if it can realize which part of the environment can be influenced by its actions. For simplicity, we use *controllable elements* to denote those environment elements that can be influenced by the agent's actions and thus reflect the action-related dynamics. We argue that these controllable elements should be explicitly retained for better representations.

Following the intuition above, we develop an efficient and flexible framework named **L**earning **C**ontrollable **E**lements oriented **R**epresentations (LCER) for reinforcement learning to capture the controllable elements of the environment. Specifically, we formally define a metric to measure the amount of controllable elements that representations capture, and then propose a surrogate objective that is derived from this metric and can make the representations encode controllable elements as

well as sufficient information. By emphasizing the importance of the controllable elements, LCER can capture more task-relevant information automatically.

We evaluate our method LCER on the DMControl Suite (Tassa et al., 2018), which has been widely used in recent years to benchmark sample efficiency (Kostrikov et al., 2020b; Hafner et al., 2019; Laskin et al.; Lee et al., 2020b). Our experiments show that LCER combined with SAC (Haarnoja et al., 2018) can outperform other representation learning methods on a majority of tasks, and even achieve better performance than the state-based SAC on the DMControl 100K benchmark. We also evaluate LCER on the Distracting Control Suite (Stone et al., 2021), showing LCER's ability to filter out distracting factors. The implementation of our work is available at `https://anonymous.4open.science/r/LCER-FF40`

## 2 RELATED WORK

In this section, we review the common approaches of representation learning in RL briefly.

**Predictive representations in RL.** Predictive representations are typically trained to predict the properties of future states. One commonly used approach is to predict the representation of future states given the current state's representation and actions. Such an approach is widely used in model-based RL algorithms. The representations are prone to collapse if without further constraints (Gelada et al., 2019), thus a reconstruction loss is usually applied to ensure non-trivial representations. However, this approach tries to encode everything, unable to filter out task-irrelevant information. SPR (Schwarzer et al., 2020) introduces a target encoder to avoid collapse and reconstruction, but its effect lacks theoretical guarantee, and it does not emphasize the controllable elements as LCER does. There are some methods that train predictive representations from information-theoretic perspectives, such as (van den Oord et al., 2018; Hjelm et al., 2019; Anand et al., 2019; Lee et al., 2020b; Mazoure et al.). These methods do not give a bias explicitly towards what kind of environment's dynamics should the representations reflect, while our method LCER focuses more on the action-related dynamics which is more likely to be task-relevant. Our experiments in Section 5.2 shows that encoding predictive information does not ensure representations of high quality in complex environments, and it is crucial to retain controllable elements explicitly.

There are some prior works that also emphasize controllable elements. Typically, (Zhang et al., 2018; pok, 2016; Badia et al., 2020) learn an inverse model that predicts the action given successive states. However, these approaches typically rely on a reconstructive loss, which will significantly harm their performance in complex environments. There are also some works that emphasize the importance of actions in representation learning by introducing various action-related objectives such as policy selectivity(Bengio et al., 2017) and contingency awareness(Bellemare et al., 2012). However, their methods are often built on heuristics and do not give a clear definition of controllable elements. Compared with these works, LCER firstly introduces a novel measure of controllable elements (Eq.(3)) and then develops an efficient method to optimize it without the aid of reconstruction loss.

**Bisimulation-based Representations in RL.** Bisimulation-based representations group those states into a cluster which are indistinguishable w.r.t. future return, given any action sequence tested. The indistinguishability of two states is typically measured by the bisimulation metrics, which are firstly introduced in (Ferns & doina Precup, 2014). Recently, (Zhang et al., 2020) proposes DBC, a representation learning algorithm based on bisimulation that presents impressive results in distracting environments. However, these methods rely on a well-defined reward function, which is probably unavailable in real-world tasks. Besides, they often fail to solve simple tasks efficiently such as ones from DMControl Suite (Zhang et al., 2020).

**Prior-knowledge-based Representations in RL** There are other methods that use prior knowledge to constrain the representations (Jonschkowski & Brock, 2015; Jonschkowski et al., 2017; Thomas et al., 2018). For example, (Jonschkowski & Brock, 2015) forces the representations to satisfy multiple priors such as proportionality and repeatability, which generally describe how the optimal representations should look like. However, such priors need to be specially designed for each environment, and much domain knowledge is required to form good representations. These drawbacks significantly limit the applications of these methods.

## 3 PRELIMINARIES

### 3.1 NOTATION

We assume the underlying environment is a Markov decision process (MDP), described by the tuple $M = (S, A, P, R, \gamma)$, where $S$ is the state space, $A$ the action space, $P : S_t \times A_t \times S_{t+1} \rightarrow [0, 1]$ the transition probability function which determines the distribution of next state given current state and action, and $\gamma \in [0, 1]$ a discount factor. Given the current state $s \in S$, an agent choose its action $a \in A$ according to a policy function $a \sim \pi(\cdot|s)$. This action will update the system state to a new state $s'$ according to the transition function $P$, and then a reward $r = R(s, a, s') \in R$ is given to the agent. The goal of the agent is to maximize the expected cumulative rewards by learning a policy $\pi$.

We denote the state at time step $t$ by $s_t$. We use the upper letter of $s$ (*i.e.* $S$) to refer to the random variable of state if there is no ambiguity. We use $s_{t:t+k}$ to present the state sequence from $t$ to $t + k$, *i.e.* $s_{t:t+k} := (s_t, s_{t+1}, ..., s_{t+k})$. These notations can be extended to other variables such as actions and rewards. We use $\phi : S \rightarrow Z$ to denote the embedding function of LCER, which maps $s \in S$ to a latent space $Z$.

### 3.2 MUTUAL INFORMATION

The mutual information(MI) of two random variables $X, Y$ is defined as:

$$I(X;Y) = D_{KL}(P_{XY}\|P_X P_Y), \tag{1}$$

where $D_{KL}$ is the Kullback–Leibler divergence that measures the difference of two distributions.

The conditional mutual information(CMI) $I(X;Y|Z)$ is defined as the expected value of $I(X;Y)$ given the value of $Z$:

$$I(X;Y|Z) = E_{z \sim Z}I(X|z;Y|z). \tag{2}$$

## 4 LEARNING CONTROLLABLE ELEMENTS ORIENTED REPRESENTATIONS

Given an embedding function: $\phi : S \rightarrow Z$, we argue that the number of controllable elements that $\phi$ contains can be measured by the conditional mutual information(CMI):

$$I_k(\phi) := I(\phi(S_{t+k}); A_{t:t+k-1}|\phi(S_t)) \tag{3}$$

Intuitively, $I_k$ measures the information that $\phi(S_{t+k})$ and $A_{t:t+k-1}$ shares given $\phi(S_t)$. Note that the non-controllable elements captured by $\phi$ do not contribute to $I_k$ because they are not related to the actions. Training representations to maximize Eq.(3) is an efficient way to make the representations capture controllable elements. However, representations obtained in this way may be too compressive[1] to provide sufficient information for solving tasks in some cases, because not all task-relevant elements are controllable. For example, in navigation tasks, the position of the goal is task-relevant but not controllable by the agent. Therefore, we need to utilize Eq.(3) in a more delicate manner. Specifically, in Section 4.1 we propose a surrogate objective derived from Eq.(3) which additionally provides a mechanism to control the degree of compression. In Section 4.2, we describe how to optimize this objective, leading to our final algorithm LCER.

### 4.1 THE OPTIMIZATION OBJECTIVE FOR LCER

By the chain rule of mutual information, we can break the $I_k$ defined in Eq.(3) into the subtraction of two terms: $I_k(\phi) = I([\phi(S_t), A_{t:t+k-1}]; \phi(S_{t+k})) - I(\phi(S_t); \phi(S_{t+k}))$. The first term tries to capture predictive information, while the second term serves as a compression term that aims to filter out action-irrelevant information. To make $\phi$ capture the controllable elements and maintain sufficient information at the same time, we introduce a hyper-parameter $\beta$ to control the degree of compression:

$$I_k(\phi; \beta) := I([\phi(S_t), A_{t:t+k-1}]; \phi(S_{t+k})) - \beta I(\phi(S_t); \phi(S_{t+k})). \tag{4}$$

---

[1]By saying "compressive", we mean that these representations only encode controllable elements and filter out any other information.

By choosing different $\beta$ when training $\phi$ to maximize Eq.(4), we can control the degree of compression, making it possible to adapt to different environments by choosing different $\beta$.

However, directly maximizing $I_k(\phi; \beta)$ is intractable, because maximizing the first term and minimizing the second term in Eq.(4) at the same time have an antagonistic effect to each other, thus it is difficult to tune in practice. In fact, we find this will often lead to unstable training process in our early experiments.

To tackle this problem, we substitute $I(\phi(S_t); \phi(S_{t+k}))$ with $I_k(\phi; \beta = 0) - I_k(\phi; \beta = 1)$:

$$
\begin{aligned}
I_k(\phi; \beta) &= I([\phi(S_t), A_{t:t+k-1}]; \phi(S_{t+k})) - \beta(I_k(\phi; \beta = 0) - I_k(\phi; \beta = 1)) \\
&= \underbrace{I([\phi(S_t), A_{t:t+k-1}]; \phi(S_{t+k}))}_{J_k^1} \cdot (1 - \beta) + \underbrace{I(\phi(S_{t+k}); A_{t:t+k-1}|\phi(S_t))}_{J_k^2} \cdot \beta.
\end{aligned}
\tag{5}
$$

In Eq.(5), we maximize two terms ($J_k^1$ and $J_k^2$) at the same time, and their coefficients are $1 - \beta$ and $\beta$ respectively. The first term $J_k^1$ is closely related to the predictive information encoded by $\phi$. The second term $J_k^2$ is related to the controllable elements, which will play an important role in distracting environments as shown in Section 5.3.

In practice, we sum over different choices of $k$, leading to the final objective to maximize:

$$
I^K(\phi; \beta) := \sum_{k=1}^{K} I_k(\phi; \beta).
\tag{6}
$$

## 4.2 How to Optimize the Objective for LCER

To maximize $I^K(\phi, \beta)$, we propose to maximize its corresponding lower bound, which is common practice in mutual information estimation. In this section, we first provides the lower bound for $J_k^1$ and $J_k^2$ respectively that are used in this paper, and finally put them together to get our final loss for training $\phi$.

**Optimizing $J_k^1$.** In order to maximize $J_k^1(\phi)$, we propose to maximize the InfoNCE (van den Oord et al., 2018) lower bound of $J_k^1(\phi)$:

$$
\hat{J}_k^1(\phi) = -\frac{1}{B} \sum_{i=1}^{B} \log \frac{\exp(f(\phi(s_t^i), a_{t:t+k-1}^i; \phi(s_{t+k}^i)))}{\sum_{j=1}^{B} \exp(f(\phi(s_t^i), a_{t:t+k-1}^i; \phi(s_{t+k}^j)))},
\tag{7}
$$

where $f$ is any scalar function, $B$ is the batch size, and $s_t^i, a_{t:t+k-1}^i, s_{t+k}^i$ is a transition sequence from $s_t^i$ to $s_{t+k}^i$. There are other alternatives to InfoNCE such as NWJ (Poole et al., 2019) and MINE (Belghazi et al., 2018), however InfoNCE can often lead to better representations as shown in (Tschannen et al., 2020). In order to efficiently optimize $\hat{J}_k^1$ for all $1 \leq k \leq K$, we recursively define $\psi_{t+k}^i$ to encode the information of $(\phi(s_t^i), a_{t:t+k-1}^i)$:

$$
\psi_{t+k}^i = \begin{cases} \phi(s_t^i), & if \ k = 0 \\ h(\psi_{t+k-1}^i, a_{t+k-1}^i), & otherwise \end{cases}
\tag{8}
$$

and set $f$ to be the bi-linear inner product of $\psi_{t+k}^i$ and $\phi(s_{t+k}^i)$:

$$
f(\phi(s_t^i), a_{t:t+k-1}^i; \phi(s_{t+k}^i)) := (\psi_{t+k}^i)^T W \phi(s_{t+k}^i).
\tag{9}
$$

Note that the $\phi_{t+k}^i$ is shared across all $\hat{J}_k^1$, thus we do not need to re-compute it. The $h$ in Eq.(8) is quite similar to a "prediction" network, but is trained together with $\phi$ instead of prediction errors.

**Optimizing $J_k^2$.** In order to maximize $J_k^2(\phi)$, we turn to maximizing a Jensen-Shannon CMI estimator Eq.(10), because it is proven to be more stable in our experiments, which is also suggested in prior works (Hjelm et al., 2019; Sanchez et al., 2020). We introduce a statistics network $T_k : S \times A_{t:t+k-1} \times S \to R$ and maximize:

$$
\hat{J}_k^2(\phi) := -\frac{1}{B} \sum_{i=1}^{B} [\log(1 + e^{-T_k(\phi_t^i, a_{t:t+k-1}^i, \phi_{t+k}^i)}) + \log(1 + e^{T_k(\phi_t^i, a_{t:t+k-1}^{j(i)}, \phi_{t+k}^i)})],
\tag{10}
$$

Table 1: The DMControl 100K benchmarks, which report the performance at 100000 environment steps in the PlaNet benchmarks. Our method LCER achieves the best performance on **6 of 6** tasks, and is the only one that outperforms StateSAC w.r.t the average returns.

| Environment | LCER(ours) | PI-SAC | SLAC | CURL | Dreamer | StateSAC |
|---|---|---|---|---|---|---|
| cartpole-swingup | $\mathbf{790 \pm 40}$ | $772 \pm 49$ | $327 \pm 44$ | $582 \pm 146$ | $234 \pm 163$ | $860 \pm 7$ |
| cheetah-run | $\mathbf{505 \pm 38}$ | $271 \pm 57$ | $413 \pm 67$ | $299 \pm 48$ | $158 \pm 133$ | $206 \pm 28$ |
| walker-walk | $\mathbf{622 \pm 127}$ | $431 \pm 80$ | $528 \pm 41$ | $403 \pm 24$ | $216 \pm 124$ | $664 \pm 111$ |
| reacher-easy | $\mathbf{704 \pm 160}$ | $687 \pm 133$ | $342 \pm 96$ | $538 \pm 233$ | $147 \pm 117$ | $774 \pm 111$ |
| finger-spin | $\mathbf{961 \pm 56}$ | $942 \pm 84$ | $951 \pm 32$ | $767 \pm 56$ | $33 \pm 42$ | $749 \pm 234$ |
| ball_in_cup-catch | $\mathbf{933 \pm 25}$ | $878 \pm 117$ | $917 \pm 21$ | $769 \pm 43$ | $172 \pm 215$ | $951 \pm 18$ |
| Average Returns | **752** | 663 | 579 | 559 | 160 | 700 |

where $j(1), .., j(B)$ are obtained by randomly shuffling $1, .., B$. The derivation of Eq.(10) is provided in Appendix B. In practice, we choose $T_k(\phi_t, a_{t:t+k-1}, \phi_{t+k}) = T(\psi_{t+k}, \phi_{t+k})$, where $\psi_{t+k}$ is a latent variable defined in Eq.(8) and $T : Z \times Z \to R$ is a network shared across different $T_k$.

**Final loss for training $\phi$.** Inspired by (He et al., 2020), we propose to utilize a *target encoder* $\phi_{targ}$ to stabilize the training process, whose weights are the exponential moving average(EMA[2]) of $\phi$. Specifically, the embeddings of $s_{t+k}^i$ ($k \geq 1$) are calculated by $\phi_{targ}$.

All pieces together, we obtain our loss for $\phi$ to minimize:

$$
\begin{aligned}
L^K(\phi; h, W, T) &= -\sum_{k=1}^{K} \hat{J}_k^1(\phi) \cdot (1-\beta) + \hat{J}_k^2(\phi) \cdot \beta \\
&= \frac{1}{B} \sum_{i=1}^{B} \sum_{k=1}^{K} [(1-\beta) \log \frac{\exp((\psi_{t+k}^i)^T W \phi_{targ}(s_{t+k}^i))}{\sum_{j=1}^{B} \exp((\psi_{t+k}^i)^T W \phi_{targ}(s_{t+k}^j))} \\
&\quad + \beta(\log(1 + \exp(-T(\psi_{t+k}^i, \phi_{targ}(s_{t+k}^i))) + \log(1 + \exp(T(\hat{\psi}_{t+k}^i, \phi_{targ}(s_{t+k}^i)))) ],
\end{aligned}
\tag{11}
$$

where $B$ is the batch size, $h, \psi_{t+k}^i$ are defined in Eq.(8), $W$ is defined in Eq.(9), $\hat{\psi}_{t+k}^i$ is a latent defined in the same way with $\psi_{t+k}^i$ but it encodes $\phi(s_t^i)$ and $a_{t:t+k-1}^{j(i)}$(instead of $a_{t:t+k-1}^i$), $T$ is a scalar function, and $\phi_{targ}$ is the target encoder. The parameters of $\phi, h, W, T$ are training together by minimizing $L^K$. In practice, Eq.(11) can be used as an *auxiliary loss* for any RL algorithms such as SAC (Haarnoja et al., 2018).

## 5 EXPERIMENTS

In this section, we first compare LCER with other methods in Section 5.1 on the PlaNet benchmark. In Section 5.2, we then demonstrate LCER's strength to filter out distracting factors on the Distracting Control Suite, and show that it is not enough to obtain good representations just by encoding predictive information. In Section 5.3, we investigate the effect of $K$ and $\beta$, which are the most important parameter in LCER. Finally, we investigate how well the representations obtained by LCER can be generalized to other tasks from the same domain in Section 5.4.

In our experiments, LCER is equipped with SAC (Haarnoja et al., 2018). Most of the parameters in LCER are the same with PI-SAC (Lee et al., 2020b), including the batch size, learning rate, action repeat, and the network architecture of $\phi$, see Appendix A for more details. Following (Laskin et al.; Lee et al., 2020a;b), we report the performance using the true environment steps, which is invariant to the choice of action repeat. Unless specially specified, We set $K = 5, \beta = 0.1$ and run every experiment using 10 random seeds.

---

[2]EMA is a widely used trick in RL. For example, in SAC, the weights of target $Q$ are the EMA of $Q$.

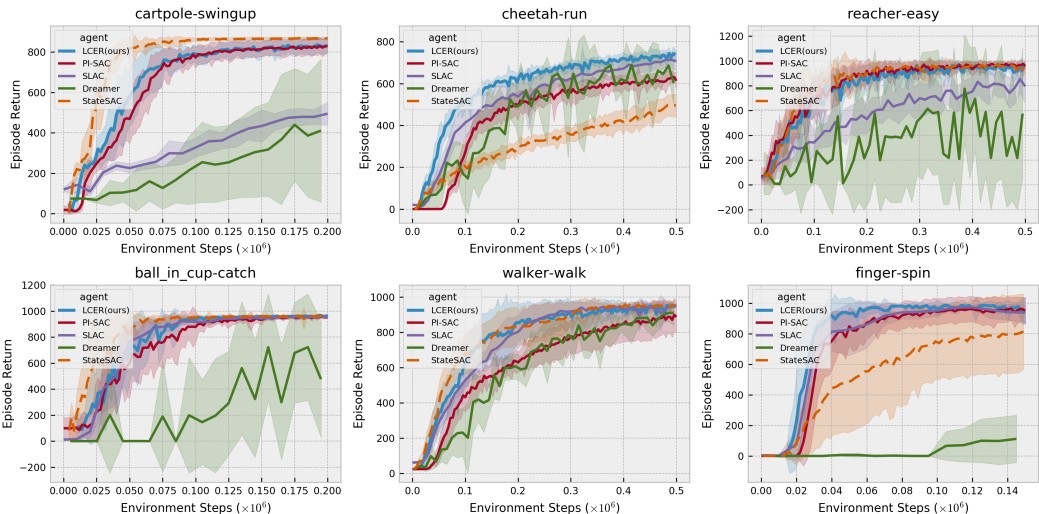

Figure 1: The PlaNet Benchmarks. The performance of StateSAC can be seen as an upper bound because it learns directly from internal physical states rather than pixels. Our algorithm LCER consistently achieves better or comparable performance than other methods(except StateSAC) on all environments

## 5.1 EVALUATION ON SAMPLE EFFICIENCY

In this section, We evaluate LCER on the PlaNet benchmark, which consists of six challenging control tasks from the DMControl Suite (Tassa et al., 2018). The PlaNet benchmark is first introduced in (Hafner et al., 2019) and later widely used to benchmark sample efficiency in (Kostrikov et al., 2020b; Laskin et al.; Lee et al., 2020b;a).

We equip LCER with SAC (Haarnoja et al., 2018) and compare LCER to 3 leading representation learning algorithms[3]: CURL (Laskin et al.),SLAC (Lee et al., 2020a),and PI-SAC (Lee et al., 2020b). CURL introduces contrastive learning into RL and actually trains representations to encode information that is invariant under data augmentation. SLAC efficiently combines both representation learning and control into a single objective via Bayesian inference, but it relies on a reconstruction loss. PI-SAC tries to encode predictive information, while LCER focuses on encoding controllable elements. All these methods are also built on SAC (Haarnoja et al., 2018). Following (Laskin et al.; Kostrikov et al., 2020b), we use the version of PI-SAC that performs one gradient update per environment step to ensure a fair comparison. We also compare LCER with a model-based algorithm Dreamer (Hafner et al., 2020). To further evaluate the quality of the representations obtained by LCER, we also consider the state-based SAC (which called StateSAC) that directly learns from internal physical states provided by DMControl Suite. The performance of StateSAC can be seen as an upper bound since it utilizes the oracle representations of the environments. All hyperparameters of StateSAC are the same with LCER except that it uses states as input instead of pixels.

In Figure 1, we report the learning curves of LCER, PI-SAC (Lee et al., 2020b), SLAC (Lee et al., 2020a), and Dreamer (Hafner et al., 2020). LCER consistently achieves better or comparable performance than other methods. In Table 1, We report the performance of each algorithm at 100k environment steps. LCER is the only algorithm that enables pixel-based SAC to outperform state-based SAC w.r.t the average returns, showing that the obtained representations can match the oracle descriptions(*i.e.* the physical states) of the environment. We also provide additional aggregate metrics on DMControl 100k benchmark, see Section C for details.

In practice, pixel-based observations are often equipped with data augmentation to serve as a regularizer (Kostrikov et al., 2020b; Lee et al., 2020b; Laskin et al.; 2020). In LCER, we also randomly shift the observations by $[-4, 4]$ pixels. However, a proper data augmentation method is not always available for general forms of observations. Therefore, We further test our method LCER in the

---

[3]The performance data of SLAC, Dreamer, CURL is provided by the authors of the corresponding papers.

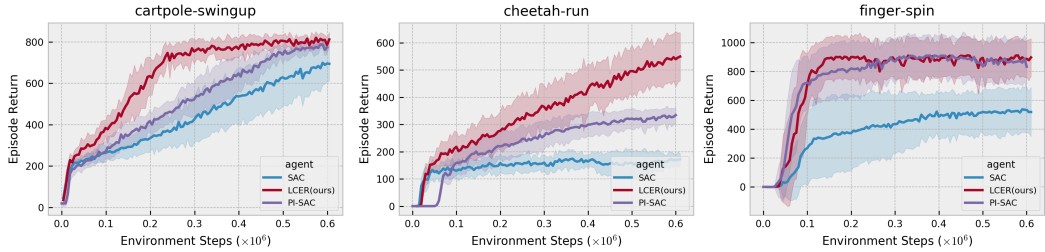

Figure 2: The learning curves when data augmentation is not available. LCER can still continuously improve the performance to a great extent without the help of data augmentation, showing LCER's ability to handle different data formats (beyond pixels) of observations.

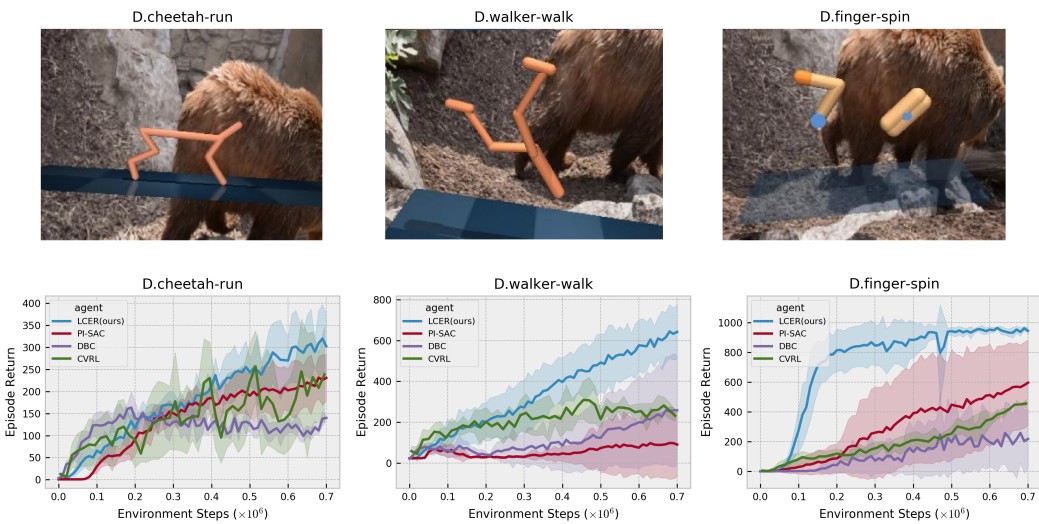

Figure 3: The Learning curves on Distracting Control Suite, a challenging benchmark that contains several kinds of visual distractions. Top row: the observations on each task. Bottom row: the learning curves on each task. The relatively poor performance of PI-SAC indicates that encoding predictive information does not ensure representations of high quality. Although these tasks are so difficult that SAC fails to solve, LCER can still improve the agent's performance gradually, showing LCER's robustness to distractions.

situation where no data augmentation is available. As shown in Figure 2, LCER can still improve the performance to a great extent without the help of data augmentation. Note that some sample-efficient RL algorithms such as DrQ (Kostrikov et al., 2020b) and RAD (Laskin et al., 2020) that rely on data augmentation can not be extended to such a situation naturally.

Our experiments on the PlaNet benchmark show that our proposed method LCER can obtain representations that help solve the tasks, and thus improve the sample efficiency.

## 5.2 ROBUSTNESS TO DISTRACTIONS

In this section, we test LCER on the Distracting Control Suite (Stone et al., 2021), a challenging benchmark that is implemented by introducing several kinds of visual distractions to the DMControl Suite, such as random changes to camera pose, object color, and background (see Figure 3). These distractions widely exist in the real world and can reduce the performance of RL algorithms to a great extent (Stone et al., 2021). We aim to show LCER's ability to filter out those distractions and that it is not enough to just encode predictive information. Note that some representation learning methods such as SLAC (Lee et al., 2020a) rely on a reconstruction loss, which will lead to significant performance degradation as shown in (Zhang et al., 2020).

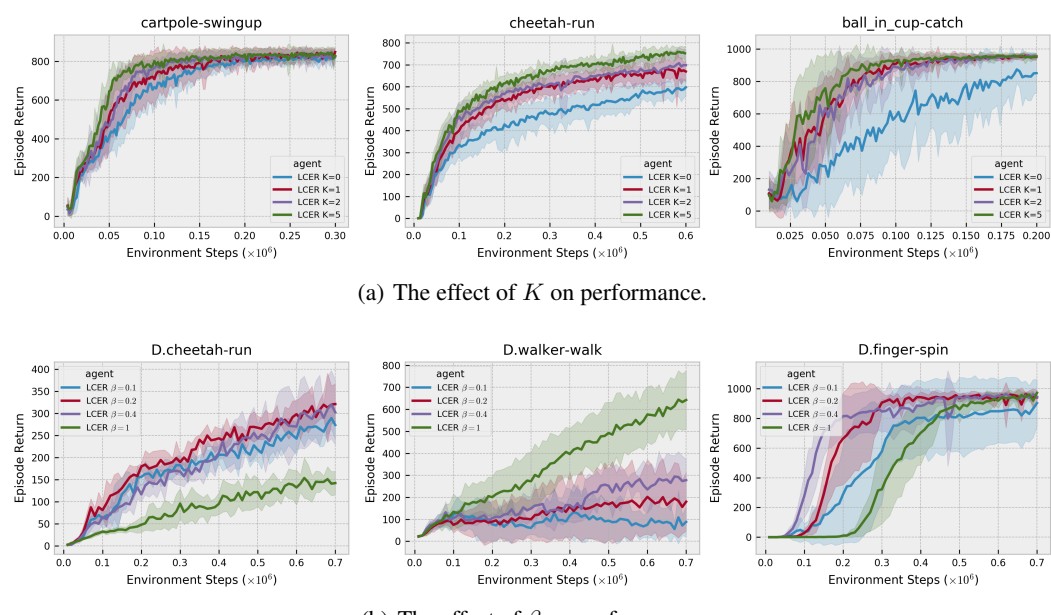

(a) The effect of $K$ on performance.

(b) The effect of $\beta$ on performance.

Figure 4: The effect of $K$ and $\beta$. Figure 4(a) shows that larger K generally leads to better performance. The effect of $\beta$ is more complicated as shown in Figure 4(b). Larger $\beta$ does lead to better performance within a certain range, however when $\beta$ goes beyond this range($i.e.\beta = 1$), it may hinder the performance. In practice, we can gradually increase the value of $\beta$ to search for optimal performance.

In Figure 3, we run experiments on three control tasks from the Distracting Control Suite: D.cheetah-run, D.walker-walk, D.finger-spin[4]. In all tasks, we set the difficulty to be 'easy' and use a dynamically varying background (refer to (Stone et al., 2021) for more details). We choose $\beta = 0.4$ for D.cheetah-run and D.finger-spin, and $\beta = 1$ for D.walker-walk[5]. We additionally compare LCER with DBC (Zhang et al., 2020) and CVRL (Ma et al., 2020), which are specially designed to deal with distractions. As shown in Figure 3, proper representation learning methods are crucial in these tasks, because it is difficult for the agent to filter out task-irrelevant information by itself. LCER achieves much better performance than PI-SAC, which indicates that it can not lead to high-quality representations just by encoding predictive information, instead we have to attach more importance to the controllable elements. We argue that this is because the distractions are often not controllable by the agent, and thus can be filtered out by LCER. Although these distractions do have a negative effect on LCER's performance, LCER suffer less than other methods, showing LCER's robustness to these distractions.

## 5.3 THE EFFECT OF $K$ AND $\beta$

In this section, we try to specify the effect of $K$ and $\beta$, which are the most important hyperparameters in LCER.

To investigate the effect of $K$, we set $\beta = 0$ and report the performance for $K = 0, 1, 2, 5$ in Figure 4(a). Note that $K = 0$ is a special case in LCER that does not take into account any actions, and exactly recovers the work of CURL (Laskin et al.). We add $K = 0$ to emphasize the importance of actions. As shown in Figure 4(a), larger $K$ leads to better performance, especially when $K$ changes from 0 to 1. We think this is because large $K$ can provide more supervision signals thanks to our special design of Eq.(11). To investigated the effect of $\beta$, we report the performance for $\beta = 0.1, 0.2, 0.4, 1$ in Figure 4(b). As shown in Figure 4(b), larger $\beta$ does lead to better performance

---

[4]We use "D." to distinguish these tasks from the ones in the DMControl Suite.

[5]We sweep over $\{0.1, 0.2, 0.4, 1\}$ to get the optimal $\beta$. The learning curves for these $\beta$'s are provided in Figure 4(b).

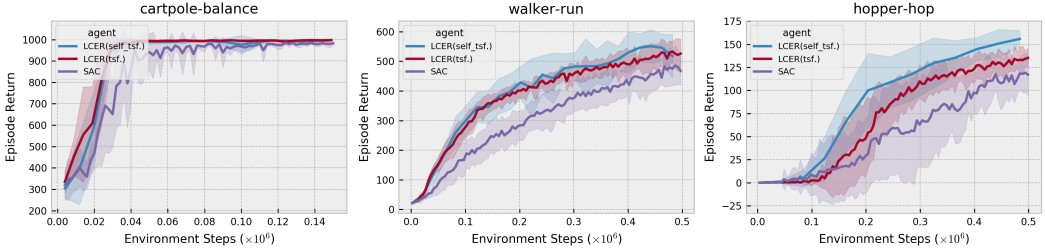

Figure 5: The learning curves of transferring obtained representations to target tasks. From left to right, the source tasks are cartpole-swingup, hopper-stand, walker-walk respectively. "LCER" and "SAC transferred" refer to using the representations obtained by LCER and vanilla SAC respectively, and "SAC" refers to training from scratch using vanilla SAC on target tasks. The representation obtained by LCER can encode more useful information than vanilla SAC, leading to better performance on target tasks.

within a certain range, however when $\beta$ goes beyond this range ($e.g.\beta = 1$), it may hinder the performance in some tasks. The result in Figure 4(b) indicates that the optimal $\beta$ should be chosen according to the given environment. By introducing a hyper-parameter $\beta$, LCER is able to strike a balance on the proportion of controllable elements that $\phi$ captures, allowing for searching for better performance.

## 5.4 THE QUALITY OF REPRESENTATIONS

In this section, we try to measure the quality of $\phi$ by generalizing $\phi$ to unseen tasks. Specifically, we train $\phi$ on a source task, froze $\phi$, and then apply it to a target task within the same domain. If LCER can capture the controllable elements efficiently, then the representations obtained on source task should also fit in with the target task, because tasks within the same domain share the same environment dynamics (but with different reward functions). In Figure 5, we report 3 learning curves on target tasks[6]: SAC, LCER(tsf.) and LCER(self_tsf.). SAC is trained directly on target tasks. LCER(tsf.) and LCER(self_tsf.) is trained with frozen encoders which are obtained via training on source tasks and target tasks respectively for 300k environment steps. Note that LCER(self_tsf.) can be seen as a ground truth baseline where an LCER agent learns representations directly on the target task and is then made to "transfer" to the same task. On those source tasks where the environment dynamics are well explored (such as cartpole-swingup and walker-walk), LCER(tsf.) can perform as well as LCER(self_tsf.) on target tasks. However, when solving target tasks needs to exploit more environment dynamics which are not explored in source tasks (such as hopper-stand[7]), LCER(tsf.) shows inferior performance. We can conclude that the transfer ability of LCER is mainly influenced by the exploration on the environment dynamics instead of task-specific features. If the agent can explore the full dynamics (such as in walker-walk and cartpole-swingup), LCER(tsf.) can learn as effective representations as LCER(self_tsf) does even on source tasks.

## 6 CONCLUSION

In this paper, we introduce the LCER, an efficient framework to train representations that can reflect the action-related dynamics of the environment. By focusing on the controllable elements of the environment, LCER can encode task-relevant information into representations, improving the performance of RL algorithms. Our experiments on DMControl 100K benchmark show that LCER can enable the pixel-based SAC to outperform state-based SAC, demonstrating improvement over prior works. We also evaluate LCER on the Distracting Control Suite, showing LCER's robustness to distractions.

---

[6]including cartpole-swingup → cartpole-balance, walker-walk → walker-run, and hopper-stand → hopper-hop.

[7]This because in hopper-hop the agent needs to move forward, however in hopper-stand it does not.

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

# A    IMPLEMENTATION

In this section, we explain the implementation in details.

**Action Repeat**    Following (Lee et al., 2020b; Laskin et al.; Lee et al., 2020a; Hafner et al., 2019), we use different action repeat to different environments. The action repeat is a parameter that determine how frequently the agent is supposed to make a decision. For a fair comparison, we adopt the action repeat from (Lee et al., 2020b;a). Specifically, for cheetah-run, cartpole-swingup, reacher-easy, and ball_in_cup-catch, we use action repeat of 4. For walker-walk and finger-spin, we use 2 and 1 respectively. Note that the action repeat parameters can directly affect the performance.

**Initial collecting step**    Following (Lee et al., 2020b;a), we pre-train our model using a small amount of random data. Specifically, we first interact with the environment using a random policy to collect initial experience, and then using the experience to pre-train our encoder $\phi$. Note that the amount of initial random data has been taken into account in all experiments. For cheetah-run, walker-walk, reacher-easy, and ball_in_cup-catch, we collect random data for 10000 true environment steps and then train $\phi$ for 10000 iterations. For finger-spin, 10000 true environment steps and 50000 iterations respectively. For cartpole-swingup, 4000 true environment steps and 1000 iterations.

**Network Architectures**    We use the encoder($i.e.$ $\phi$) that shares the same architecture with (Laskin et al.; Lee et al., 2020b), which consists of 4 convolution layers(32 3×3 filters with stride 2 for the first layer, and 32 3×3 with stride 1 for the rest 3 layers) and a fully-connected layer that flattens the output of convolution layers and then maps it into a latent space with 50 dimensions. For the $h$ in Eq.(8), we use MLPs with 2 128-d hidden layers, whose input and output are both of 50 dimensions. For the actor and critic, we both use MLPs with two 512-d hidden layers. We use tanh to squash the output of the actor to make sure it fits the action bound of the environment.

**Other Hyperparameters**    We list the other hyperparamters in Table 2. Most of these paramters are the same with (Lee et al., 2020b).

| Hyperparameter | Value |
|---|---|
| Batch size | 256 |
| Learning rate | 3e-4 |
| Replay buffer size | 100000 |
| Dimension of latent space | 50 |
| Stacked frames | 3 |
| Optimizer | Adam |
| $(\beta_1, \beta_2)$ in Adam | (0.5, 0.999) for $\alpha$, (0.9, 0.999) for others |
| target update interval | 1 |
| actor log std | [-10, 2] |
| Q EMA $\tau_i$ | 0.005 |
| Encoder EMA $\tau_\phi$ | 0.05 |
| Initial temperature | 0.1 |
| Discount | 0.99 |
| Non-linearity | ReLU |
| Entropy Target | -dim(A) |

Table 2: The hyperparameters of LCER.

# B ESTIMATION OF CONDITIONAL MUTUAL INFORMATION

In this section, we describe how we optimize $I^{JSD}$ defined in Section 4.2. In the followings, the $\phi(S_{t+k}), A_{t:t+k-1}, \phi(S_t)$ are denoted by $X, Y, Z$ respectively.

We first borrow some theoretical results from (Nowozin et al., 2016):

$$
\begin{aligned}
I^{JSD}(X;Y|Z) &= E_{z \sim Z} D_{JS}(P_{XY|z} \| P_{X|z} P_{Y|z}) \\
&= D_{JSD}(P_{XYZ} \| P_{XY} P_{Y|Z}) \\
&\geq \sup_V E_{x,y,z \sim XYZ} V(x,y,z) - E_{x,z \sim XZ, y \sim Y|z} f_{JSD}^*(V(x,y,z)).
\end{aligned}
\tag{12}
$$

Here, $f_{JSD}$ is the generator function of Jensen-Shannon divergence: $f_{JSD}(u) = -(u+1) \log \frac{1+u}{2} + u \log u$, and $f_{JSD}^*$ is its corresponding convex conjugate function: $f_{JSD}^*(t) = \sup_u \{ut - f_{JSD}(u)\} = -\log(2 - \exp(t))$. We choose $V(x,y,z) = \log(2) - \log(1 + \exp(-T_k(x,y,z)))$, then:

$$
\begin{aligned}
I^{JSD}(X;Y|Z) &\geq \sup_V E_{x,y,z \sim XYZ} V(x,y,z) - E_{x,z \sim XZ, y \sim Y|z} f_{JSD}^*(V(x,y,z)) \\
&= \sup_{T_k} -E_{x,y,z \sim XYZ} \log(1 + e^{-T_k(x,y,z)}) - E_{x,z \sim XZ, y \sim Y|z} \log(1 + e^{T_k(x,y,z)}) + 2 \log 2 \\
&\approx \sup_{T_k} -E_{x,y,z \sim XYZ} \log(1 + e^{-T_k(x,y,z)}) - E_{x,z \sim XZ, y \sim Y} \log(1 + e^{T_k(x,y,z)}) + 2 \log 2.
\end{aligned}
\tag{13}
$$

In Eq.(13), we suppose $p(y|z) \approx p(y)$(i.e. $A_{t:t+k-1}$ is independent from $\phi(S_t)$) for simplicity. Such an assumption is appropriate in our case, because $(S_t, A_{t:t+k-1}, S_{t+k})$ is sampled from a large replay buffer($= 10^5$) whose action policy is quite non-stationary due to policy updates.

To estimate the RHS of Eq.(13), we first randomly sample $B$ transitions $\{s_t^i, a_{t:t+k-1}^i, s_{t+k}^i\}_{i=1}^B$ from the replay buffer. Assume $(j(1), .., j(B))$ is a permutation of $(1, 2, .., B)$ obtained by random shuffling, then $E_{x,y,z \sim XYZ}(\cdot)$ can be approximated by $\frac{1}{B} \sum_{s_t^i, a_{t:t+k-1}^i, s_{t+k}^i}(\cdot)$ and $E_{x,z \sim XZ, y \sim Y}(\cdot)$ by $\frac{1}{B} \sum_{s_t^i, a_{t:t+k-1}^{j(i)}, s_{t+k}^i}(\cdot)$. Therefore, the RHS of Eq.(13) can be estimated by:

$$
\hat{J}_k^2(\phi) := -\frac{1}{B} \sum_{i=1}^B [\log(1 + e^{-T_k(\phi_t^i, a_{t:t+k-1}^i, \phi_{t+k}^i)}) + \log(1 + e^{T_k(\phi_t^i, a_{t:t+k-1}^{j(i)}, \phi_{t+k}^i)})],
\tag{14}
$$

Here, we omit the "$2 \log 2$" in Eq.(13) for simplicity.

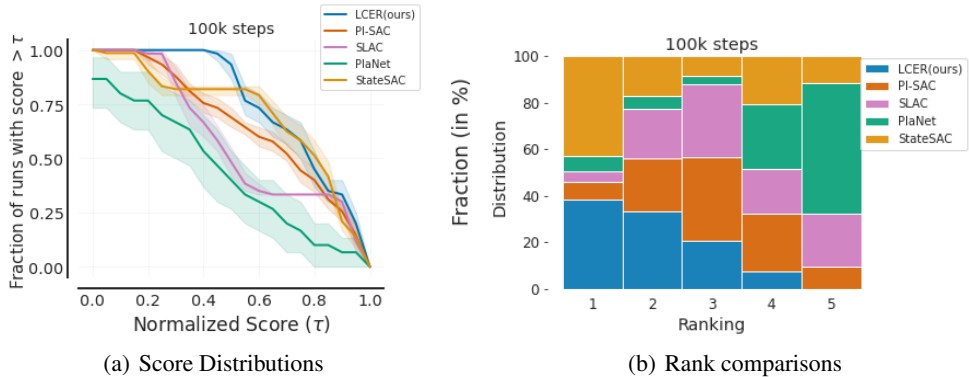

(a) Score Distributions                    (b) Rank comparisons

Figure 6: Additional aggregate metrics on DMControl 100K benchmark.

## C  ADDITIONAL AGGREGATE METRICS ON DMCONTROL 100K BENCHMARK

In this section, we provides additional comparison on DMControl 100k benchmark using aggregate metrics from $rliable$ (Agarwal et al., 2021b).

We normalize all scores to $[0, 1]$ by dividing by the maximum score (=1000). The metrics are shown in Figure 6, which include:

- Ranks distribution, where the $i^{th}$ column in the plot shows the probability that a given method is assigned rank $i$, averaged across all tasks. We can see that LCER and StateSAC shares (almost) same probability (about 40%) of ranking $1^{st}$, while LCER has a higher probability (35% v.s. 18%) of ranking $2^{rd}$

- Score distribution, where a point $(\tau, f_\tau)$ in the plot means the fraction of runs with score $> \tau$ is $f_\tau$. The curve of LCER is above StateSAC's when $\tau < 0.5$, showing than LCER are more stable than StateSAC (regarding the worst performance).

