# OpenReview forum: "Learning Controllable Elements Oriented Representations for Reinforcement Learning "
_ICLR.cc/2022/Conference — ICLR 2022 Submitted_

### Official Review · Reviewer_WDLZ · 2021-10-27

**Correctness:** 3
**Technical Novelty And Significance:** 3
**Empirical Novelty And Significance:** 3
**Recommendation:** 5
**Confidence:** 4

**Main Review:**

**Strengths:**
- paper was clearly written and well-motivated
- approach can be applied to any task
- clear mathematical exposition of the loss function used
- demonstration of better sample complexity against other baselines including CURL, PlaNet, SLAC, and PI-SAC.
- demonstration of robustness to distractors compared to a PI-SLAC baseline
- the authors have explored the robustness of the algorithms to the two new hyperparameters introduced, $\beta$ and $K$


**Comments:**
I do like the idea of using controllability to inform compression of state representations -- after all, what is important to the agent is what it can control about the environment. Yet there may be situations where this approach can be too compressive. Perhaps the most important situation is where there are non-controllable task-informative features. The authors do point out one example, the goal location. But there are in principle many others. For instance, the directions for solving a task (e.g. a map showing the solution to a maze) may be painted on a wall in the environment -- the agent could still solve the task by completely ignoring the directions, but it would do much better if it could exploit that information. Another limitation with this approach is if the environment dynamics change. For instance, suppose an agent wanted to navigate a drone to a goal. Controllability will differ depending on wind conditions. So if the agent learned a representation under e.g. no wind, that representation may be less effective if the agent had to navigate in the presence of strong and volatile wind. These limitations are by no means reasons to disqualify the paper, but it would be worth pointing this out in their discussion of their algorithm.


**Weaknesses:**
- the idea of using mutual info to guide representation learning has been considered previously in various guises. Perhaps the closest version to the current approach is by Lee et al, 2020, which uses mutual information between the initial and final states of a trajectory to guide state compression. The authors' main modification is really to introduce action sequences into the picture, motivated by the idea of using controllability to guide compression, and in this reviewer's view, this contribution seems borderline incremental. Indeed, the improvement over PI-SAC (Table 1) seems to be within statistical error for several of the tasks (cartpole swing-up, reacher easy, finger spin, and ball-in-cup), which makes me wonder to what extent is the improvement attributable to implementational differences instead.
- mutual information between two random variables depends on their probability distributions. One of the variables in their objective is the action sequence, but this depends, in particular, on the policy learnt (training data are sampled from the replay buffer). This begs the question to what extent are the representations learnt optimised for the policy (and hence the particular task). It may be helpful to freeze the encoder and evaluate the representations on different tasks in the same environment (for instance, by changing the reward function to encourage different behaviour). In this setting, what is controllable about the environment would remain the same, but the optimal policy for the task will have to differ. This would also shed light on how transferable these representations would be to different tasks in the same environment. Moreover, it would be helpful to compare against a ground truth baseline where a LCER agent learns representations directly on the target task and is then made to "transfer" to the same task -- the idea is to see how well the agent that transfers representations compares to one that has access to the "ground truth" representation.
- Section 5.4 presenting the evaluation of representation transfer was especially weak. The difference between the source and target task was not made clear, making it difficult to interpret what transfer is taking place between tasks. Moreover, why did the authors only compare against SAC baselines in this section? Clearly SAC was not designed to extract out task irrelevant features, so this seems like a weak baseline to compare against.
- as mentioned above, to what extent are these representations robust to perturbations in the state dynamics.
- a limitation of the approach is the introduction of extra hyperparameters $\beta$ and $K$, which need to be tuned for each task, particularly the weighting hyperparameter $\beta$.
- minor typographical errors including spurious capitalizations and places where the authors wrote "dose" where they meant "does" (e.g. in the middle of the 1st paragraph of the "Predictive representations in RL" subsection and in the middle of the captions for Figs 3 and 4)


**Summary Of The Paper:**

There has been a school thought lately in RL that good representations for control should focus on abstracting out on only the task-relevant features. In high-dimensional tasks, this is especially useful in making optimal use of a limited neural network capacity and may even help the agent be robust to distractors in the environment. The authors motivate an approach to learning task-relevant representations by having the representations capture only the controllable elements of the environment. Their approach is based on a mutual information loss between actions sequences and final states, and they validate their approach on a series of DMControl task where both high-dimensional pixel observations as well as ground truth latent state representations are available. The idea is to compare the performance of agents that attempt to learn representations of latent states from pixels and compare against an agent that has direct access to the ground truth latent state. In their empirical results they show that:
- their new approach (called LCER) is able to outperform alternative approaches on a set of these tasks. On some tasks, it even outperforms an SAC agent with access to the ground truth states.
- LCER is able to outperform alternatives when distracting visual features are introduced into these tasks
- LCER's representations learned on a source task are able to transfer better to downstream tasks.

**Summary Of The Review:**

While I liked the motivation of their approach, my main hesitation for recommending this paper lies both on the strength of their contribution and issues with their empirical results. On the strength of their contribution, to me it seems borderline incremental, so I would be curious to hear the opinion of the other reviewers. Moreover, the results seem marginally better than PI-SAC, which makes me wonder to what extent the better results can the better results be conclusively attributed to the better objective and not to implementational factors, like the choice of $h()$ function architecture or a better approximation of the mutual information objective.

On the empirical side, their explanation of transfer in Section 5.4 could have been clearer, and they could have compared against stronger baselines. I would also have appreciated insights into how much the representations are fine-tuned to the policy being used. Finally, it would strengthen the paper if the authors could provide insight into how well do these representations transfer to different objectives as well as to perturbations of dynamics in the same environment. The main motivation behind the authors' work is to have an agent learn state representations that capture controllable elements of the environment only. Evaluating transfer on different goals in the same environment would be a test of this, since an agent that truly captured the controllable features should not be adversely affected by a change in goal or reward function.

---

> ### Author Response · Authors · 2021-11-17
> **Response (1/2)**
>
> We appreciate your constructive suggestions on further improvement of our work. We would like to address your concerns one by one. Any further discussion will be appreciated.
>
> **Q1** The authors' main modification is really to introduce action sequences into the picture, motivated by the idea of using controllability to guide compression, and in this reviewer's view, this contribution seems borderline incremental.
>
> **A1** The main difference between PI-SAC and LCER is that PI-SAC tries to encode predictive information (PI), while LCER focuses on encoding controllable elements (CE).
> Since CE belongs to PI, LCER can be seen as a variant of PI-SAC that explicitly biases the representations towards retaining CE. However, we argue that such a bias is not trivial, especially in the distracting settings in which the distractions are predictive but not controllable (e.g. we can predict the changed background in the next frame given the current background, however, the background itself is not controllable).
>
> LCER's decisive advantages over PI-SAC mainly lie in the distracting settings (as shown in Section 5.2), which are out of PI-SAC's capacity. The tasks in Figure.1 and Table.1 are relatively easy to solve, and thus have low distinguish degrees. Besides, in these tasks, the PI is almost equal to CE, which may account for the relatively marginal performance improvement.
>
> **Q2** It may be helpful to freeze the encoder and evaluate the representations on different tasks in the same environment (for instance, by changing the reward function to encourage different behavior).
>
> **A2** Yes, indeed. Actually, this is exactly what we want to explore in Section 5.4. In Section 5.4, the source task and target task shares the same dynamics, but with different reward function (cartpole-swingup->cartpole-balance, hopper-stand->hopper-hop, walker-walk->walker-run).
>
> **Q3** Moreover, it would be helpful to compare against a ground truth baseline where an LCER agent learns representations directly on the target task and is then made to "transfer" to the same task.
>
> **A3** We think this is an interesting setting and here we provide such a baseline to compare with. We name such a baseline as "LCER self_tsf". (see https://anonymous.4open.science/r/ICLR_Rebuttal-5EB3/figure5.png)
>
> We run experiments on 3 source-target tasks pair: cartpole-swingup $\rightarrow$ cartpole-balance, walker-walk $\rightarrow$ walker-run, and hopper-stand $\rightarrow$ hopper-hop. We report 3 learning curves on target tasks: SAC, LCER(tsf.), and LCER(self\_tsf.). SAC is trained directly on target tasks. LCER(tsf.) and LCER(self\_tsf.) is trained with frozen encoders which are obtained via training on source tasks and target tasks respectively for 300k environment steps. On those source tasks where the environment dynamics are well explored (such as cartpole-swingup and walker-walk), LCER(tsf.) can perform as well as LCER(self\_tsf.) on target tasks. However, when solving target tasks needs to exploit more environment dynamics which are not explored  in source tasks (such as hopper-stand. This because in hopper-hop the agent needs to move forward, however in hopper-stand it does not.), LCER(tsf.) shows inferior performance. We can conclude that the transferability of LCER is mainly influenced by the exploration on the environment dynamics instead of task-specific features. If the agent can explore the full dynamics (such as in walker-walk and cartpole-swingup), LCER(tsf.) can learn as effective representations as LCER(self\_tsf) does even on source tasks.
>
> **Q4**  The difference between the source and target task was not made clear, making it difficult to interpret what transfer is taking place between tasks.
>
> **A4** We omit the detailed difference comparison mainly because it can be inferred from the task name (cartpole-swingup->cartpole-balance, walker-walk->walker-run, hopper-stand->hopper-run). The source tasks and the target tasks lie in the same domain and thus share the same environment dynamics (but with different reward functions).
>
> **Q5** Moreover, why did the authors only compare against SAC baselines in this section? Clearly, SAC was not designed to extract out task irrelevant features, so this seems like a weak baseline to compare against.
>
> **A5** Experiments in Section 5.4 are designed as ablations aiming to investigate how well the representations obtained by LCER can be transferred to new tasks with the same dynamics but different rewards. We will add a baseline "LCER self_tsf" (described in A3) to show that if the environment's dynamics are well explored on the source tasks, LCER can learn as effective representations as "LCER self_tsf" does.

---

> > ### Author Response · Authors · 2021-11-17
> > **Response(2/2)**
> >
> > **Q6** as mentioned above, to what extent are these representations robust to perturbations in the state dynamics.
> >
> > **A6** Due to time constraints, we can not provide such an experiment. We would like to leave it for future work.
> >
> > **Q7** a limitation of the approach is the introduction of extra hyperparameters $\beta$ and $K$, which need to be tuned for each task, particularly the weighting hyperparameter $\beta$.
> >
> > **A7** Yes, we agree. BTW, the $K$ is actually easy to tune because larger $K$ is usually better, as shown in Section 5.3.
> >
> > **Q8** minor typographical errors including spurious capitalizations and places where the authors wrote "dose" where they meant "does" (e.g. in the middle of the 1st paragraph of the "Predictive representations in RL" subsection and in the middle of the captions for Figs 3 and 4)
> >
> > **A8** Thanks for your corrections.

---

### Official Review · Reviewer_uWv6 · 2021-10-29

**Correctness:** 3
**Technical Novelty And Significance:** 3
**Empirical Novelty And Significance:** 2
**Recommendation:** 5
**Confidence:** 4

**Main Review:**

## Strengths
- This paper is well structured and easy to follow.
- The empirical results in 100k DM Control benchmark (6 tasks), and even distracted-background settings, show LCER performs best among some image-based RL algorithms, which successfully reveals that learned representation of LCER captures the controllable features in the image observations.

## Weaknesses
- I think the derivation of LCER seems a bit misleading. The objective that LCER actually optimizes may not be introduced from the CMI directly; it starts with CMI between future-state embedding and action sequences given past-state embedding (Eq. 3), introduces the bottleneck parameter $\beta$ (similar to Lee et al. 2020) (Eq. 4), and deformulates the objective again (Eq. 5). In Eq. 5, $J_k^2$ is the same term as Eq. 3, which might be confusing. It can be simply argued that LCER introduces the CMI term into CURL objective ($J_k^1$) with some ratio $\beta>0$, and also introduces the surrogate form of CMI based on JSD (Eq. 10). In addition, balancing $\beta$ seems very important hyper-parameters for LCER (I guess even in standard DM Control, where $\beta=0.1$ is used. If LCER learns controllable elements from CMI, $\beta$ should be 1, but isn't).
- I am also a bit confused about the connection and difference between LCER and PI-SAC, since both of them optimize the CMI objective as an auxiliary loss for RL. While the authors just say, "PI-SAC tries to encode predictive information, while LCER focuses on encoding controllable elements", further clarification about this is required to emphasize the contribution of this paper.
- For PI-SAC baselines, the authors can include the "gradient\_step=2, 4" results for a fair comparison, since it improves the performance according to its codebase (https://github.com/google-research/pisac). In contrast, LCER can also utilize such extra gradient steps. Moreover, I find that while LCER uses $K=5$ sequence length for InfoNCE + CMI objective, PI-SAC only uses $K=3$ (from Appendix A of Lee et al. 2020), which might be "unfair" comparisons, considering the experiments excepting multiple gradient steps of PI-SAC.
- I think the authors may also include the results of Dreamer (Hafner et al. 2020) for 100k benchmark since it generally achieves better results than PlaNet; in addition, CURL and PI-SAC compared against it in their original papers (+ code is available online).
- In distracting-environments settings, the authors may also include Contrastive Variational  Reinforcement Learning (CVRL) as a baseline, proposed by Ma et al. (2020), which achieves better results compared to Dreamer, image-based SAC, and image-based D4PG (+ code is available online). In addition, Ma et al. (2020) also take a contrastive approach to handle distracting background, so it should be added as related work.


### Reference
Lee et al. Predictive Information Accelerates Learning in RL. Advances in Neural Information Processing Systems (2020).

Hafner et al. Dream to Control: Learning Behaviors by Latent Imagination. International Conference on Learning Representations (2020).

Ma et al. Contrastive Variational Reinforcement Learning for Complex Observations. Conference on Robot Learning (2020).


**Summary Of The Paper:**

This paper proposes a novel representation learning flamework, LCER, which enables the agent to extract task-relevant representations from image-based observations. It optimizes the surrogate loss of conditional mutual information (CMI) between future-state embedding and action sequences given past-state embedding. The key contribution seems the balancing between InfoNCE objective (similar to CURL) and JSD-based CMI. The experimental comparison shows that LCER performs best among some image-based RL algorithms (PI-SAC, SLAC. CURL, PlaNet); in 100k DM Control benchmark (6 tasks), and distracted-background settings.

**Summary Of The Review:**

As discussed in Main Review in detail, I think the derivation of LCER seems a bit misleading, and further clarification compared to PI-SAC is required (since both PI-SAC and LCER optimize the CMI objective). In addition, some important baselines (PI-SAC w/ gs=2 or gs=4, Dreamer, CVRL) seem missing in the current manuscript. Considering these aspects, I larn towards rejection.

---

> ### Author Response · Authors · 2021-11-17
> **Response**
>
> We are encouraged that you find our paper well structured and easy to follow. We are glad to answer your questions and would appreciate any further responses.
>
> **Q1** the derivation of LCER seems a bit misleading.
>
> **A1** The reason why we present the derivation of LCER in such a way is that we want to give readers a clear idea about what $\beta$ stands for (i.e. it actually is the coefficient of a compression term). Directly introducing LCER as a combination of MI and CMI is okay, but in this way, we can not answer the question about why MI and CMI are combinable and what the meaning of $\beta$ is. We also want to present readers with our whole thinking progress, not just what to do but also why. Besides, we have already pointed out that Eq.(5) is actually a combination of two terms in the paragraph followed Eq.(5).
>
> **Q2** the connection and difference between LCER and PI-SAC since both of them optimize the CMI objective as an auxiliary loss for RL
>
> **A2** The main difference between PI-SAC and LCER is that PI-SAC tries to encode predictive information (PI), while LCER focuses on encoding controllable elements (CE).
> Since CE belongs to PI, LCER can be seen as a variant of PI-SAC that explicitly biases the representations towards retaining CE. However, we argue that such a bias is not trivial, especially in the distracting settings in which the distractions are predictive but not controllable (e.g. we can predict the changed background in the next frame given the current background, however, the background itself is not controllable). In fact, the relation between PI and CE has already been discussed in Section 1 and Section 2.
>
> LCER's decisive advantages over PI-SAC mainly lie in the distracting settings (in Section 5.2), which are out of PI-SAC's capacity.
>
> **Q3** For PI-SAC baselines, the authors can include the "gradient_step=2, 4" results for a fair comparison, since it improves the performance according to its codebase
>
> **A3** We do not compare PI-SAC with gs=2 because other baselines(CURL, SLAC, StateSAC)  are all using gs=1. Besides, it should be a common practice to set gs=1 (e.g. DrQ[1] also compares against a gs=1 version of previous work when faced with the similar situation).
>
> In fact, the performance of LCER-gs1 is already marginally better than PI-SAC-gs2, even though LCER-gs1 only uses gs=1. Since the tasks in the PlaNet benchmarks are relatively easy to solve, both LCER-gs1 and PISAC-gs2 are already convergent on 3 (out of 6) tasks in 100k environment steps (cartpole-swingup, finger-spin, ball_in_cup-catch). On the remaining tasks, LCER-gs1 achieves better performance than PI-SAC-gs2 on 2 tasks (505 v.s. 460 on cheetah-rin, 622 v.s.514 on walker-walk). Only on reacher-easy LCER is worse than PI-SAC (704 v.s. 758).
>
> Besides, LCER's decisive advantages over PI-SAC mainly lie in the distracting settings (as shown in Section 5.2). Here we provide the comparison between LCER-gs1 and PI-SAC-gs2 on the D.cheetah-run, D.walker-walk, and D.finger-spin. (see https://anonymous.4open.science/r/ICLR_Rebuttal-5EB3/pisac_gs2.png)
> As shown in this figure, LCER-gs1 performs better than PI-SAC-gs2, even with fewer gradient steps.
>
> **Q4** I think the authors may also include the results of Dreamer (Hafner et al. 2020) for the 100k benchmark since it generally achieves better results than PlaNet;
>
> **A4** Thanks for your advice. We have replaced PlaNet with Dreamer in our latest revision.
>
> **Q5** Ma et al. (2020) also take a contrastive approach to handle distracting background, so it should be added as related work.
>
> **A5**  We have added CVRL as our baseline in our latest revision (see https://anonymous.4open.science/r/ICLR_Rebuttal-5EB3/figure3.png). In addition, we also include DBC which is a representation learning method that is designed to deal with distractions. LCER still gets better performance than these baselines.
>
> [1] Image Augmentation Is All You Need: Regularizing Deep Reinforcement Learning from Pixels

---

> > ### Comment · Reviewer_uWv6 · 2021-11-24
> > **Re: Response**
> >
> > I appreciate the authors to responding the review and providing additional experimental results I suggested, which definitely helps to improve the paper.
> >
> > **> Comment to A1, A2**
> >
> > I've been aware of the conceptual difference between PI and CE (CE $\in$ PI ), and what I expected to be clarified is the difference in the objective function or random variables that MI considers between PI-SAC and LCER. After I read both papers again, I found that PI-SAC and LCER take a different approach to learn representations. I concern that the performance gap in distracting environments came from such embedding choices rather than PI-CE difference (similar discussion in https://arxiv.org/abs/2106.07278).
> >
> > **> Comment to A3**
> >
> > I don't strongly request the author to include the results of PISAC-gs2, but I note that I pointed out the difference of sequence length between PI-SAC (K=3) and LCER (K=5). As far as I observed in Figure 4 (a), the longer K is better.
> >
> > Related to the above (Comment to A1, A2), I also have a concern about the effectiveness of LCER in distracting settings. Figure 3 and Figure 4 (b) show that in D.cheetah_run, LCER takes small $\beta$ and PI-SAC is comparable to LCER, which implies PI is enough to control the cheetah. In D.walker_walk, LCER takes large $\beta$ and PI-SAC is inferior to LCER (and the same as small-$\beta$-LCER), which implies CE is effective to control the walker. These results follow the intuition that $\beta$ controls the degree of PI-CE trade-off. However, in D.finger_spin, $\beta=1$ is similar to PI-SAC, and small $\beta$ improves a lot. It is hard to understand this intuitively. Could the authors explain this behavior?
> > In addition, I concern that while LCER argues the advantage in distracting settings,  the evaluation on such settings is not so extensive as normal settings (3 for distracting, 6 for normal). Transferability of learned representation is also tested in only normal settigs. For example, CVRL is tested on 10 DM control tasks for both distracting and normal settings. Also, due to the embedding and implementation difference between PI-SAC and LCER, $\beta=0$ baseline might be important.

---

> > > ### Author Response · Authors · 2021-11-25
> > > **2nd Response**
> > >
> > > We appreciate the reviewer to provide additional feedback.
> > >
> > > **Q1** I concern that the performance gap in distracting environments came from such embedding choices rather than the PI-CE difference
> > >
> > > **A1** In conclusion, there are 2 differences between PI-SAC and LCER: (D1) embedding choices, architecture designs, etc, and (D2) PI-CE difference. We think both (D1) and (D2) are advantages of LCER. To investigate (D2) separately, we can compare LCER with LCER($\beta=0.1$) which can be seen as a variant of PI-SAC with the same embedding choices as LCER. As shown in Fig.4, LCER can still achieve better performance than LCER($\beta=0.1$) by choosing different $\beta$, which indicates that (D2) does contribute to the performance gap.
> > >
> > > **Q2** the difference of sequence length between PI-SAC (K=3) and LCER (K=5). As far as I observed in Figure 4 (a), the longer K is better.
> > >
> > > **A2** We feel sorry that we forget to discuss the choice of K in our initial response. Although we have no experiments to prove this, we think PI-SAC can not leverage longer K as efficiently as LCER can due to the architecture difference between PI-SAC and LCER. This is because PI-SAC utilize a 'one-step' forward/backward encoder that takes the whole $A_{t:t+k}$ as input, whereas LCER use a 'transition model'-like $h$ that allows for aggregating additional supervision signals from $S_{t+1},..,S_{t+k-1}$. Overall, we think it is another advantage of LCER that LCER can make better use of longer K.
> > >
> > > **Q3** However, in D.finger_spin, $\beta=1$ is similar to PI-SAC, and small $\beta$  improves a lot. It is hard to understand this intuitively.
> > >
> > > **A3** The intuition about the optimal $\beta$ is that we should choose $\beta$ to be the **minimal** $\beta$ that is sufficient to encode most CEs. The dynamics of D.walker-walk are much more complicated than D.finger-spin (In walker-walk we need 24-dim physical states to describe the observation, whereas in D.finger-spin it is only 9-dim. Besides, the action space is 6-dim and 2-dim respectively). Therefore CE in D.walker-walk is more difficult to capture than D.finger-spin, thus we need higher compression ability (i.e. bigger $\beta$) on D.walker-walk. However, on the D.finger-spin smaller $\beta$(i.e. $\beta=0.4$) is enough to encode CE, and can also allow capturing more information from PI which is potentially task-relevant.
> > >
> > > Note that the curve of $\beta=1$ on D.finger-spin is still growing faster than PI-SAC (though it spends more env steps on initial exploration and training $\phi$). Btw, we think the performance improvement of $\beta=0.1$ over PI-SAC is mainly because of (D1).
> > >
> > > **Q4** while LCER argues the advantage in distracting settings, the evaluation on such settings is not so extensive as normal settings
> > >
> > > **A4** This is mainly because conducting extensive experiments in DC(i.e. Distracting Control suite) is much more difficult than that in the DMC(i.e. DMControl suite) due to the slow running speed of DC. In fact, the running speed of DC is approximately 10x slower than DMC, which will significantly increase the total training time for each trial. In addition, DC is not a common benchmark, thus we can not get the learning curves of baselines from the corresponding authors directly.
> > >
> > > In addition, we compare 3 baselines (except the vanilla SAC) in the distracting settings, whereas CVRL only compares 1 baseline (except the vanilla SAC). We think the number of baselines can compensate for the relative lack of extensive tasks.
> > >
> > > **Q5** Also, due to the embedding and implementation difference between PI-SAC and LCER,  baseline $\beta=0$ might be important.
> > >
> > > **A5** Thanks for your advice, we will try to add $\beta=0$ as a baseline. In fact, in our preliminary experiments, we found that the performance of $\beta=0$ is much close to $\beta=0.1$ in the distracting settings.

---

### Official Review · Reviewer_pqPt · 2021-11-02

**Correctness:** 2
**Technical Novelty And Significance:** 3
**Empirical Novelty And Significance:** 2
**Recommendation:** 5
**Confidence:** 4

**Main Review:**

## Strengths

1.  The paper is generally well written and well motivated. While section 4 required multiple passes (likely due to multiple approximations that are used), the high-level motivation and ideas are easy to grasp.

2. I like the idea - it is quite simple and elegant - learn representations that retain information about the controllable elements of the environment.

3. I liked the experiments and ablations that the paper includes. While I have several issues with the claims/interpretation-of-results, I think the experiment design is well thought.

## Areas for improvement

1. Several claims are not well-substantiated.

    * In some cases, it could just be a matter of rewording. For example "LCER is the first representation learning algorithm that enables the pixel-based SAC to outperform state-based SAC on the DMControl100K benchmark". From Table 1, LCER lags behind StateSAC on 5 (out of 6 environments). While its performance is impressive, the claim itself is not supported. Comparing the average performance (across 6 envs) is misleading because of 1 outlier environment.

    * In other cases, the claims are not tested. For example, there is no experiment to show that the representations indeed "retain the controllable elements of the environment" (which is the key motivation behind the work).

    * In several cases, the results are quite close to each other (eg Figure 1 walker walk, reacher-easy, ball-in-cup, cartpole...) In absence of statistical significance test, it is difficult to conclude that the proposed approach is indeed better. Same applies to several results in the tables as well.

2. When evaluation robustness to distractors, better baselines (like Zhang et al 2020) should be used.

## Further questions

Note that I did not consider these points as weakness as I did not fully understand these aspects.

1. In equation 4, what is the meaning of "$I([\phi(S_t), A_{t:t+k-1}]$" ie what is the "[...]" operator doing?
2. In equation 3, is only A conditioned on $\phi(S_t)$ or both $\phi(S_{t+k})$ and $A$ are conditioned?
3. In page 3, Section 4, the authors correctly point out "For example, in navigation tasks, the position of the goal
is task-relevant but not controllable by the agent" as a motivation for controlling the degree of compression. However, their experiments did not have any goal conditioned environments. In that case, would it make sense to directly use equation 3 (atleast in the context of experiments considered in the paper).
4. In page 4, the authors mentions "To tackle this problem, we substitute I..." (right before equation 5). Could they describe why is this approximation valid?
5. What is the runtime cost of LCER compared to the baseline? I expect it to be much higher than the baseline (given it is used k-step lookaheads to compute the losses. Reporting the runtime comparison will be useful for a holistic comparison. Similarly, I expect LCER to be using much more params and would like to see a comparison of those numbers.
6. What is the effect of using the target encoder? How severe is effect if target encoder isnt used?
7. In page 6, the paper mentions "All parameters of StateSAC.." do you mean all hyper-parameters of StateSAC... ?
8. In figure 2, cartpole results seems to be stopped too early. The two baselines may be able to overtake the red curve.
9. In page 7, the paper mentions "In LCER, we also randomly shift the observations by [−4, 4] pixels". Is this aimed to provide translation invariance? If yes, isnt a conv net already providing that? Is this augmentation applied to the baselines as well?

**Summary Of The Paper:**

The paper proposes the LCER algorithm that trains representations to retain the controllable elements of the environment, which can reflect the action-related environment dynamics and thus are likely to be task-relevant. The main experiments use DMControl suite and includes several ablation experiments.


**Summary Of The Review:**

The paper proposes a simple and elegant idea and shows that the approach "works" in practice. However, it is difficult to evaluate the effectiveness/significance of the approach. I look forward to the author's response and interacting with them to understand their approach better.

---

> ### Public Comment · ~Rishabh_Agarwal2 · 2021-11-09
> **Suggestion for improving statistical reliability of results**
>
> We found that on DM control (100k and 500k benchmark), there was substantial overlap in 95% CIs of reported mean scores [1]. ​For reliable evaluation with a few runs, one possible suggestion for the authors could be to report the average probability of improvement (and other metrics like mean/IQM) with confidence intervals. Score distributions might also be useful for showing variation in results across different tasks and runs.  The authors can easily do so using the library at https://github.com/google-research/rliable or the [colab](https://bit.ly/statistical_precipice_colab).
>
> Also, note that the individual runs for DMC 100k/500k benchmark (provided by the authors of the corresponding papers) are at https://console.cloud.google.com/storage/browser/rl-benchmark-data/dm_control for other statistical analysis.
>
> [1] Agarwal, R., Schwarzer, M., Castro, P.S., Courville, A. and Bellemare, M.G., 2021. Deep reinforcement learning at the edge of the statistical precipice. In NeurIPS.

---

> ### Author Response · Authors · 2021-11-17
> **Response (1/2)**
>
> We are encouraged that you find our idea simple and elegant. We are glad to answer your questions and would appreciate any further response.
>
> **Areas for improvement**
>
> **Q1** From Table 1, LCER lags behind StateSAC on 5 (out of 6 environments). While its performance is impressive, the claim itself is not supported. Comparing the average performance (across 6 environments) is misleading because of 1 outlier environment.
>
> **A1** Actually LCER outperforms StateSAC on 2 (out of 6) tasks in Table 1. If we take Figure 1 into account, we can conclude that LCER achieves better performance than StateSAC on 2 tasks(cheetah-run and finger-spin), comparable performance on 2 tasks(walker-walk and reacher-easy), and worse performance also on 2 tasks(cartpole-swingup, ball\_in\_cup-catch). This means that LCER actually is comparable to StateSAC at least, thus we think it is safe to use 'average return' to claim 'LCER outperforms StateSAC' on 100k benchmark.
>
> We provide other evidence that supports our claim in our latest revision. These evaluation metrics are proposed in [1].
> - (1) Aggregate ranks distribution (see https://anonymous.4open.science/r/ICLR_Rebuttal-5EB3/ranks.png), where the $i^{th}$ column in the plot shows the probability that a given method is assigned rank $i$, averaged across all tasks. We can see that LCER and StateSAC shares (almost) same probability (about 40%) of ranking $1^{st}$, while LCER has a higher probability (35% v.s. 18%) of ranking $2^{rd}$.
>
>
> - (2) Score distribution (see https://anonymous.4open.science/r/ICLR_Rebuttal-5EB3/score_distribution.png), where a point $(\tau, f_\tau)$ in the plot means the fraction of runs with score $> \tau$ is $f_\tau$. The curve of LCER is above StateSAC's when $\tau < 0.5$, showing than LCER are more stable than StateSAC (regarding the worst performance).
> ![](figures/score_distribution_100k-1.png)
>
>
> **Q2**  There is no experiment to show that the representations indeed "retain the controllable elements of the environment" (which is the key motivation behind the work).
>
> **A2**
> By definition, Eq(3) is a metric that measures the amount of CE(i.e. controllable elements). Therefore, LCER directly maximizes the amount of CE by maximizing Eq.(3).
>
> To give more intuitive and direct evidence that the representations indeed "retain the CE of the environment", we report curves showing that the amount of CE indeed increases during the training process (see https://anonymous.4open.science/r/ICLR_Rebuttal-5EB3/cmi.PNG). In this figure, we estimate the value of CMI (i.e. Eq.(3), which is a measure of CE) using a CMI estimator NWJ. We train this estimator until convergence independently for different periods (i.e. across different environment steps) in the training process. A point $(x,y)$ on the curve means that the representations obtained in environment step $x$ can retain about $y$ CEs.
>
>
> In fact, retaining controllable elements is the key feature of LCER that accounts for the considerable performance improvement on the Distracting Control Suite, given the fact PI-SAC which encodes predictive information only performs relatively poorly on these tasks.
>
> **Q3** In several cases, the results are quite close to each other (eg Figure 1 walker walk, reacher-easy, ball-in-cup, cartpole...) In absence of a statistical significance test, it is difficult to conclude that the proposed approach is indeed better. The same applies to several results in the tables as well.
>
> **A4** Actually these 6 tasks in the DMControl Suite are too easy to solve in that there are no distractions in these environments (thus the predictive information is almost equal to CE). This is why LCER's performance in these tasks seems close to PI-SAC.
>
> However, We emphasize that LCER's decisive advantages mainly lie in the distracting settings (as shown in Section 5.2). These tasks are more close to real-world applications. In these tasks, the varying backgrounds are also predictive (since we can predict the background in the future given the current background), but is task-irrelevant. Only controllable elements are task-relevant in these scenarios. PI-SAC will encode these task-irrelevant elements, whereas LCER will exclude them. This may probably account for the performance difference between LCER and PI-SAC, and can also serve as circumstantial evidence that LCER indeed retains controllable elements.
>
> **Q4** When evaluating robustness to distractors, better baselines (like Zhang et al 2020) should be used.
>
> **A4** Thanks for your advice. We have included DBC (Zhang et al 2020) as well as CVRL[2] as baselines in our latest revision. CVRL is a leading model-based method that aims to deal with distractions too. LCER still achieves the best performance in these tasks.

---

> > ### Author Response · Authors · 2021-11-17
> > **Response (2/2)**
> >
> > **Further questions**
> >
> > **Q5** In equation 4, what is the meaning of " $I([A,B];C)$" ie what is the "[...]" operator doing?
> >
> > **A5** $[A,B]$ means that we treat $Y=[A,B]$ as a random variable whose distribution function is the joint distribution function of $A$ and $B$.
> >
> > **Q6** In equation 3, is only $A$ conditioned on $\phi(S_k)$ or both $\phi(S_{t+k})$ and $A$ are conditioned?
> >
> > **A6** Both.
> >
> > **Q7** However, their experiments did not have any goal-conditioned environments. In that case, would it make sense to directly use equation 3 (at least in the context of experiments considered in the paper).
> >
> > **A7**
> > It depends. If the reward signals are dense and informative enough, then it is possible that directly optimizing Eq.(3) can still get good representations, because Eq.(3) is used as auxiliary loss in SAC. However, if the rewards are sparse, we think optimizing Eq.(3) can hinder the performance because it will filter out important information such as the goal locations. This is why we introduce $\beta$ to control the compression degree.
> >
> > **Q8** On page 4, the authors mention "To tackle this problem, we substitute I..." (right before equation 5). Could they describe why is this approximation valid?
> >
> > **A8** Actually, this is not an approximation. Instead, it is a direct application of the chain rule of mutual information.
> >
> > **Q9** What is the runtime cost of LCER compared to the baseline? I expect it to be much higher than the baseline (given it is used k-step lookaheads to compute the losses. Reporting the runtime comparison will be useful for a holistic comparison. Similarly, I expect LCER to be using much more params and would like to see a comparison of those numbers.
> >
> > **A9** Here are the runtime costs for different methods (over 300k environment steps, on V100):
> >
> > |SAC | CURL  |LCER      | PI-SAC |
> > |-|-| ----------- | ----------- |
> > |3.2 hours |5.75 hours|6.25 hours|9.5 hours|
> >
> > Actually, LCER does not consume much more time than other representation learning methods. This is because the main time-consuming component is the forward/backward pass of the encoder. Although LCER introduces k-step lookaheads, these additional $k$ forward passes of the encoder are highly parallel and thus can be computed efficiently by GPU.
> >
> > The additional params introduced by LCER are $h,W,T$, which have approximately 30k, 2.5k, 12.8k parameters respectively. Thus the total is about 45k. This is a relatively small number, given the fact the amount of parameters in SAC is about 640k.
> >
> > **Q10** What is the effect of using the target encoder? How severe is effect if target encoder isnt used?
> >
> > **A10** Target encoder is especially important in contrastive learning. In a word, it can prevent the encoder from trivial solutions. We encourage the reviewers to refer to [3] for more details.
> >
> > **Q11** On page 6, the paper mentions "All parameters of StateSAC.." do you mean all hyper-parameters of StateSAC... ?
> >
> > **A11** Yes. Thanks for your correction.
> >
> > **Q12** In figure 2, cartpole results seem to be stopped too early. The two baselines may be able to overtake the red curve.
> >
> > **A11** In fact, the highest achievable score of cartpole in these methods is about 800+(see Figure1), which has already been achieved by the red curve.
> >
> > **Q12** On page 7, the paper mentions "In LCER, we also randomly shift the observations by [−4, 4] pixels". Is this aimed to provide translation invariance? If yes, isnt a conv net already providing that? Is this augmentation applied to the baselines as well?
> >
> > **A12** We think this is to guarantee the agent captures the invariant information under translation. Conv net does provide such invariance, but it's not enough in RL. Previous work (DrQ, RAD, CURL,etc) shows that applying such a data augmentation is crucial in pixel-based tasks. In contrastive learning, it has also been shown that using random shifts can lead to better representations. Such augmentation is applied to all baselines except PlaNet and SLAC, because it may hinder their(PlaNet, SLAC) performance, which has been shown in SLAC's paper. This is probably because they rely on a reconstruction loss, which will force the representations to recover the randomly shifted images.
> >
> > [1] Agarwal, R., Schwarzer, M., Castro, P.S., Courville, A. and Bellemare, M.G., 2021. Deep reinforcement learning at the edge of the statistical precipice. In NeurIPS.
> >
> > [2] Ma et al. Contrastive Variational Reinforcement Learning for Complex Observations. Conference on Robot Learning (2020).
> >
> > [3] Tian et al. Understanding Self-Supervised Learning Dynamics without Contrastive Pairs https://arxiv.org/pdf/2102.06810.pdf
> >
> > [4] Stochastic latent actor-critic: Deep reinforcement learning with a latent variable model

---

> > ### Comment · Reviewer_pqPt · 2021-12-01
> > **Thanks for the response!**
> >
> > I thank the authors for responding to my questions/feedback. I agree with much of the reply. I have one more question
> >
> >  > LCER does not consume much more time than other representation learning methods. This is because the main time-consuming component is the forward/backward pass of the encoder. Although LCER introduces k-step lookaheads, these additional  forward passes of the encoder are highly parallel and thus can be computed efficiently by GPU.
> >
> > Arent the k steps sequential ? How can they be performed in parallel ?
> >
> > I will be updating my score post the discussion with the other reviewers.

---

> > > ### Author Response · Authors · 2021-12-01
> > > **Response**
> > >
> > > Thanks for your response!
> > >
> > > Although the calculations of $\psi_{t+k}, k=1,2..$ are sequential (and thus cannot be parallelized), they are actually carried out in a small latent space (about 50-dim). Therefore, they are not as time-consuming as the forward passes of the encoder (i.e. the calculation of $\phi(S_t)$). In fact, calculating $\phi$ takes approximately about $\geq$8M FLOPs (only multiplication operations counted), whereas $\psi_{t+k},k=1,2,..$ only 0.15 M FLOPs totally.

---

### Official Review · Reviewer_vvtm · 2021-11-02

**Correctness:** 3
**Technical Novelty And Significance:** 3
**Empirical Novelty And Significance:** 3
**Recommendation:** 6
**Confidence:** 3

**Main Review:**

The high-level idea seems sound and practical and the implementation strategy builds on popular ideas in the literature and seems reasonable.

Does not cite DIYAN (Eysenbach 2018) https://arxiv.org/abs/1802.06070 which also uses mutual info between latent state projection Z and actions A, though not for representation learning.

In equation 6, the information in the sequence is given as the sum of information at each horizon value k which is probably a sensible approximation, but I am guessing not technically correct as the states at each horizon value are not mutually independent.

Page 5 middle paragraph starting “where B is the batch size” contains “is a latent defined” which I am guessing should be “is a latent variable defined”.

Why is JSD more stable than MI? Is it possible to give an intuition?

I didn’t find the section on augmentation directly relevant to the paper’s point and I think it could be omitted from the main text if space was tight. It might allow the paper to expand section 4.2 a bit more to explain the recursion better.

Alternative: LACER: Learning Agent Controllable Elements Representations

**Summary Of The Paper:**

The paper proposes a method (LCER) for agents to develop compact task representations from high-dimensional inputs (e.g., images) during unsupervised exploration that are effective for downstream tasks. The paper hypothesizes that encouraging an agent’s representation to focus on changes related to the agent’s actions will be more useful to downstream tasks than representations that try to naively model all changes. In order to allow some non-controllable elements in the representation the paper proposes a blended objective which trades off representing action controllable aspects of state with non-controllable elements of state using an application specific empirically tuned hyperparameter \beta. The paper is set in a fully observable world where a state at time t, S_{t}, is projected to an instantaneous latent state Z_{t}. The idea of controllable elements is implemented by computing mutual information between the latent state projection in the future, Z_{t+k} and the actions taken by the agent I(  Z_{t+k}; A_{t:t+k-1} ).  The uncontrollable information is uncontrollable elements is implemented by computing mutual information between current and future latent states I( Z_{t}; Z_{t+k} ).  Mutual information between latent state and actions uses the neural InfoNCE bound trick. For the second term, the paper instead optimizes Jensen-Shannon divergence between current and future latent representations instead of mutual information, again using a neural bound as the paper claims this is more stable. The paper then shows that this objective can be combined with a standard soft actor-critic architecture to learn representations without extrinsic rewards. In empirical studies on the PlaNet, the paper’s proposed LCER method finds representations that can be used by a policy learner to converge faster than competing methods. The proposed LCER method also compares well to an SAC agent that has access to underlying state representation of the domain (e.g., joint angles, velocities). A second set of experiments on the Distracting  Control Suite which has a lot of extraneous background noise show wider separation between LCER and competing methods. A study shows that increasing the look ahead K improves the quality of representations.

**Summary Of The Review:**

I think the idea of explicitly trading off the action focused vs. state focused loss to generate representations from unsupervised experience is new but closely related to precedents. It speeds convergence on easier PlaNet tasks and seems empirically to make a difference in achievable expected reward for the harder tasks in the Distracting Control Suite.

---

> ### Author Response · Authors · 2021-11-17
> **Response**
>
> Thanks for your thorough summary of our work and we are encouraged that you have carefully read our paper. We would like to discuss your concerns one by one. Any further discussion will be appreciated.
>
> **Q1** why not cite DIYAN
>
> **A1** Although both DIYAN and LCER develop their own training objective through information perspectives, DIYAN focuses on a relatively different problem.  Therefore, we tend not to cite it.
>
> **Q2** In equation 6, the information in the sequence is given as the sum of information at each horizon value k
>
> **A2** In fact, the sum of $I_k$ in Eq.6 is just a trick to provide more supervision signals (instead of serving as an approximation of information in the sequence). Applying Eq.6 can also improve the robustness of our method when choosing different k.
>
> **Q3** ...contains “is a latent defined”...:
>
> **A3** Thanks for fixing the bug, we will correct it.
>
> **Q4** Why is JSD more stable than MI? Is it possible to give an intuition?
>
> **A4** By saying JSD is more stable than MI, we actually mean that the $D_{JS}$-based CMI estimator (Eq.10) is more stable than other $D_{KL}$-based CMI estimator (such as MINE, NWJ, etc). (In fact, Eq.10 can also be seen as an estimator for $D_{KL}$-based CMI, because the value of $D_{JS}$-based and of $D_{KL}$-based CMI should have an almost linear relationship (see the appendix in https://openreview.net/pdf?id=Bklr3j0cKX)).
>
> The $D_{JS}$-based CMI estimator Eq.10 is actually a binary cross-entropy, which trains a classifier $T_k$ to distinguish positive and negative samples. Such a classifier should be easier to train and can provide a more reasonable backpropagation gradient to the encoder.
>
> To avoid any possible misleading, we re-state the reason why we choose Eq.10 in our paper:
>
> In order to maximize $J^2_k(\phi)$, we turn to maximize a Jensen-Shannon MI estimator Eq.1(0), because it is proven to be more stable in our experiments, which is also suggested in prior works.
>
> **Q5**  the section on augmentation
>
> **A5** we think the no-data-augmentation setting is essential in that it can reveal LCER's adaptability to different data formats. However, we are willing to move it into the appendix as long as there emerges more interesting and insightful content that can be added into our paper.
>
> **Q6** I think the idea of explicitly trading off the action focused vs. state focused loss to generate representations from unsupervised experience is new but closely related to precedents.
>
> **A6** The closest previous work of LCER is PI-SAC. We would like to make clear the difference between PI-SAC and LCER here. The main difference between PI-SAC and LCER is that PI-SAC tries to encode predictive information (PI), while LCER focuses on encoding controllable elements (CE). Since CE belongs to PI, LCER can be seen as a variant of PI-SAC that explicitly biases the representations towards retaining CE. However, we argue that such a bias is not trivial, especially in the distracting settings in which the distractions are predictive but not controllable.

---

### Author Response · Authors · 2021-11-17
**Change List**

After considering the suggestions from reviewers,  we have published a new revision of our paper.

Here we summarize the main changes:
- Dreamer is added as a baseline in Section 5.1, according to the suggestion of reviewer uWv6.
- A new section is added in the Appendix to provide more aggregate metrics on DMControl 100K benchmark.
- More baselines are added in Section 5.2 (in the distracting settings), including DBC and CVRL
- A ground truth baseline (i.e. LCER(self\_tsf.)) is added in Section 5.4, according to the suggestion of reviewer WDLZ.

---

### Decision · Program_Chairs · 2022-01-20

**Decision:**

Reject

**Comment:**

This paper introduces an objective for representation learning that captures "controllable elements" in the environment (i.e., things that are affected by the agent's actions). In their reviews and discussion, the reviewers agreed this idea was intuitive, well-motivated, and the paper well written. However, multiple reviewers raised concerns about the evaluation and the extent to which LCER is truly an improvement over PI-SAC. Although many of the reviewer's concerns were addressed in the rebuttal period, at the end of the discussion they were still unconvinced or confused about how much LCER really helps over PI-SAC. Based on this, my assessment is that this paper is a promising piece of work, and that with some more controlled comparisons (see suggestion below) it would be a useful contribution to the literature. However, given that the claims are not fully supported as it currently stands, I recommend rejection.

Specific suggestion to improve the paper: based on reading the paper and the discussion, it seems to me (as per the authors' own statement in response to Reviewer uWv6) that the most valid/controlled comparison between LCER and PI-SAC is in Figure 4, where LCER w/ $\beta=0.1$ "can be seen as a variant of PI-SAC with the same embedding choices as LCER" (author's words). However, when taking into account the error bars of the training curves, other values of $\beta$ are only clearly better than $\beta=0.1$ in 1/3 environments (D.walker-walk). This does not make for a particularly convincing result that LCER is better than PI-SAC. To improve the paper, I'd encourage the authors to run further well-controlled comparisons such as this in a larger number of environments. If they can show via such controlled comparisons that LCER is generally better than PI-SAC (i.e. LCER w/ $\beta=0.1$) then that would be a much more compelling demonstration of LCER's superiority.